# The clinical demand and supply of blood in India: A National level estimation study

**Joy John Mammen** [1]*, **Edwin Sam Asirvatham** [2], **Jeyaseelan Lakshmanan** [3], **Charishma Jones Sarman** [2], **Arvind Pandey** [4], **Varsha Ranjan** [2], **Bimal Charles** [2], **Thenmozhi Mani** [1], **Sunil D Khaparde** [5], **Sunita Upadhyaya** [6], **Shobini Rajan** [7]

1 Christian Medical College, Vellore, Tamil Nadu, India, 2 Christian Medical Association of India (CMAI), New Delhi, India, 3 Professor of Biostatistics, College of Medicine, Mohammed Bin Rashid University of Medicine and Health Sciences, DHCC, Dubai, United Arab Emirates, 4 National Institute of Medical Statistics, Ministry of Health and Family Welfare, New Delhi, India, 5 Ministry of Health and Family Welfare, Mumbai, Maharashtra, India, 6 Division of Global HIV and TB, Center for Centers for Disease Control and Prevention (CDC), New Delhi, India, 7 National AIDS Control Organization (NACO), Janpath, New Delhi, India

* joymammen@cmcvellore.ac.in

## Abstract

### Background

Estimating the clinical demand for blood and components arising in a health facility is crucial to ensure timely availability of blood. This study aims to estimate disease-specific clinical demand, supply and utilization of whole blood and components in India.

### Methods

We conducted a national level cross-sectional study in five randomly selected states from five regions of the country. We included 251 public and private facilities representing primary, secondary and tertiary care facilities. We collected annual disease-specific demand, supply and utilization of blood and components using a structured tool. We estimated the national demand by extrapolating the study data (demand and beds) to the total number of estimated beds in the country.

### Findings

According to the study, the total clinical demand of 251 health facilities with 51,562 beds was 474,627 whole blood units. Based on this, the clinical demand for India was estimated at 14·6 million whole blood units (95 CI: 14·59–14·62), an equivalent of 36·3 donations per 1,000 eligible populations, which will address whole blood and component requirement. The medicine specialty accounted for 6·0 million units (41·2%), followed by surgery 4·1 million (27·9%), obstetrics and gynecology 3·3 million (22·4%) and pediatrics 1·2 million (8·5%) units. The supply was 93% which is equivalent to 33·8 donations against the demand.

### Conclusion

The study indicated a demand and supply gap of 2.5 donations per 1,000 eligible persons which is around one million units. The gap emphasises the need for sustained and

**Data Availability Statement:** All relevant data are within the manuscript.

**Funding:** This project has been supported by the President's Emergency Plan for AIDS Relief

(PEPFAR) through the Centers for Disease Control and Prevention (CDC) under the terms of cooperative agreement number 5U2G GH001103-02.

**Competing interests:** The authors have declared that no competing interests exist.

concerted efforts from all stakeholders and for increasing the awareness about repeat voluntary non-remunerated blood donation (VNRBD); optimizing the availability of blood components through efficient blood component separation units; promoting modern principles of patient blood management and strengthening capacities of human resources in the blood transfusion system in India.

## Introduction

Blood transfusion permits increasingly complex medical and surgical interventions to significantly improve the life expectancy and quality of life of patients [1]. The timely availability of safe blood is essential to address the clinical demand that arises in health care facilities to ensure appropriate treatment and minimise preventable mortality.

Clinical demand is the total number of units of whole blood and components required to meet all blood transfusions for emergencies and elective procedures at health facilities over a defined period. The current supply is the actual supply to healthcare facilities against the demand, and utilization is the actual utilization of the supplied blood by the healthcare facilities over a defined period [2]. Apart from the social, economic, geographical, and cultural factors, clinical demand is dependent on healthcare providers' training, behavior and blood ordering practice within the health system capacity [3]. Ideally, a blood transfusion system should be capable of addressing 100% of clinical demand arising in healthcare facilities, within its catchment area. Yet, in many developing and under-developed countries, there is a widespread shortfall between demand and supply of blood due to several barriers. The major factors are increasing requirement for blood and blood products, poor implementation of voluntary donation and blood safety programs in countries, inadequate voluntary non-remunerated blood donation (VNRBD), suboptimal component separation, inadequate infrastructure, equipment and trained human resources, inappropriate use of blood and blood components, poor quality management systems, poor supply chain management systems, lack of cold chain, wastages and expiry of blood [1, 4, 5].

The blood transfusion service in India is fragmented with a network of 2,760 blood banks owned by the public, private and not-for-profit sectors, collecting around 12 million units in a year. Most (77%) blood banks were attached to hospitals and 22% were stand-alone. Around 51% had a component separation facility, separating 53% of the total annual collection in the country [6]. Despite the significant increase in the availability and use of components, whole blood is still requested and transfused substantially in clinical settings in India. As the majority of the health care facilities do not have blood banks on their premises, they are dependent on nearby blood banks or blood storage centres as available. Moreover, the proportion of voluntary blood donation is still around 80% and the remaining is depending on replacement donation by families or professionals in the guise of replacement donors. The National Blood Transfusion Council (NBTC) has regulated the cost of blood, both at public and private facilities. However, timely access to safe blood is still a challenge in many parts of the country which requires an efficient blood transfusion system. A comprehensive estimation of clinical demand, supply and utilization is critical to inform evidence-based blood donation and safety programs and strategies towards achieving universal access to blood. Challenged with a large population of approximately 1.4 billion, this study aims to estimate disease-specific clinical demand, supply and utilization of whole blood and blood components in India.

## Materials and methods

### Study and sampling design

We conducted a pilot study in Karnataka state, which was not included in the main study, to evaluate the feasibility of conducting the study, test the research protocols, data collection tools, sampling strategies, and estimate a statistically significant sample size for the study.

This study was a national level cross-sectional study conducted in five randomly selected states, one state each from five regions of the country, that are north, east, west, south and northeast (Fig 1). We randomly selected the required number of facilities from the list of primary, secondary and tertiary health care facilities representing the public and private sector, from the state specific list of health care facilities. We provisionally estimated the mean demand per bed as 9·6 (Standard deviation (SD: 6) units per annum through a pilot study. We had three strata- primary, secondary and tertiary care facilities. To estimate the demand of 9·6 (SD:6) units per bed with a precision of 0.1 units with 95% Confidence Interval (CI), we needed to study 13,830 units per stratum (primary, secondary and tertiary) which was rounded off to 15,000 units, totalling to 45,000 units of blood. Accounting for a design effect of two, we needed to study 90,000 units of blood per region, for which 9,000 beds were to be studied. We followed the proportional allocation method to allocate the samples to the strata. We considered 10%, 25% and 65% of primary, secondary and tertiary care beds respectively to transfuse blood, based on consensus among experts through four Delphi exercises conducted among 59 health care providers from across the country. We considered the average number of beds in primary, secondary and tertiary care facilities at 35, 120 and 1,000 and the presence of private

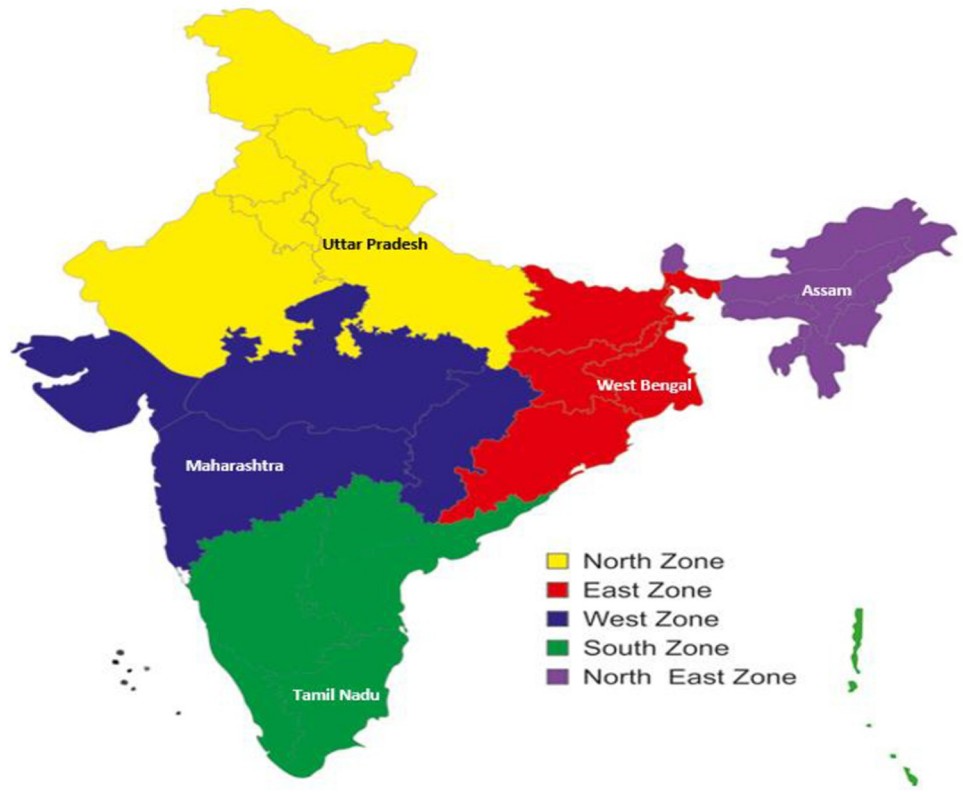

**Fig 1. Study implementation region and states.** *States—**Uttar Pradesh, West Bengal, Maharashtra, Tamil Nadu and Assam**.

and public facilities in India at the ratio of 60:40 [7]. Based on these considerations and to ensure 9,000 beds in each region, we decided to study 50 facilities from each state, of which 29 and 21 facilities were from private (58%) and public (42%) respectively. In total, we included 251 health care facilities in our study (Fig 2).

The sampling frame consists of the list of facilities from the selected states from each region. From the public health care system, we included Community Health Centers (CHCs) as primary; sub-divisional (SDH) and district hospitals (DH) as secondary and medical colleges

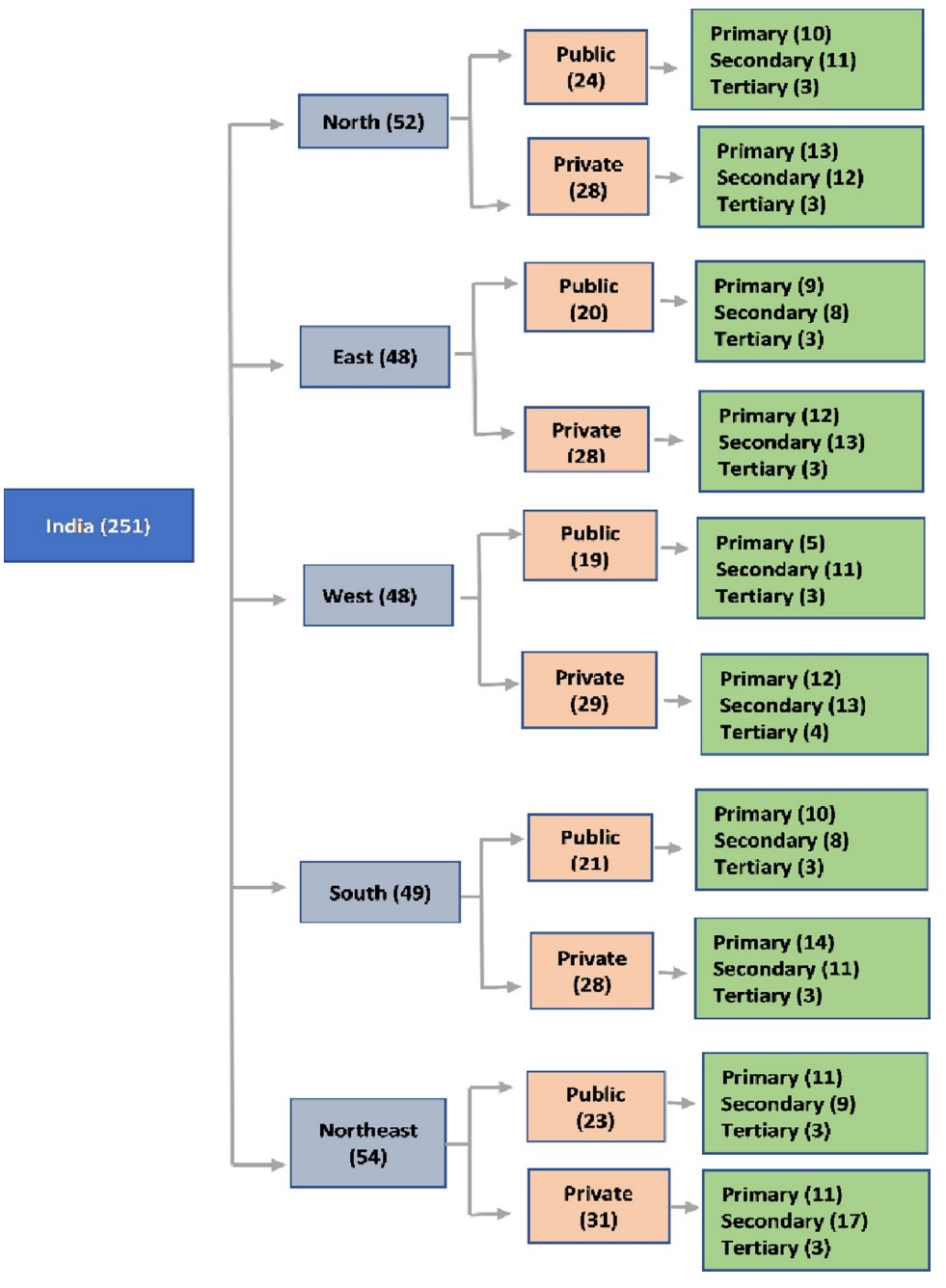

**Fig 2. Sampling strategy.**

(MC) as tertiary care hospitals. In the private sector, we included facilities providing basic medical, surgical, Obstetrics & Gynecology (O&G) and pediatrics with less than 50 beds as primary facilities providing basic and specialty services with more than 50 beds as secondary, and facilities providing basic, specialty and super specialty services including medical colleges with more than 200 beds as tertiary facilities. We collected the lists of public and private health facilities from multiple sources and triangulated them to arrive at a state-wise list.

## Data collection

We collected annual disease-specific demand, supply and utilization of blood and components from the participating health care facilities using a structured data collection form. The data included administrative details such as location, type of facility (public/private), level of care, number of beds, number of outpatients, admissions, bed occupancy rate, the average length of stay, and number of staff. The transfusion-related details included the number of patients with a disease or condition, number of patients requiring transfusion, number of units required per patient, percentage of blood supplied, utilized, discarded and/or returned to the blood bank under medicine, surgery, obstetrics and gynecology and pediatrics. We obtained these data primarily from the hospital and blood bank records manually. We conducted interviews with the heads of the administration and senior clinicians from these departments to validate the data and obtain additional information.

## Data analysis and computation of clinical demand

The data were entered in REDCap (Research Electronic Data Capture), a secure web application for building and managing surveys [8, 9]. It was extracted as SPSS file and analysed using SPSS version 24 (IBM, Armonk, NY) [10]. As a first step, we computed disease-specific demand for each institution by summing the demand for whole blood and the demand for the highest component among red cells (RBC), plasma (FFP), platelets and cryoprecipitate because one unit of whole blood can provide one unit each of all components. Besides, it is based on 350 ml per donation which is generally practiced in India. Secondly, we computed the institution-wise demand for each department i.e., medical, surgical, obstetrics and gynecology and pediatrics, by calculating all the disease-specific demand of each department. Finally, we summed up the demand of all four departments to calculate the total demand for the institution. We calculated the actual supply and utilization using the percentage of supply and utilization provided by each institution.

We estimated the national demand by extrapolating the study data (number of beds and clinical demand in the study) to the total number of estimated beds in the country. As per the National Health Profile (2017) [11], India had 634,879 beds in the public sector, which is approximately 40% of the total beds. The remaining 60% was computed at 952,319 beds in the private sector. We used the sum of the public and private beds, which is 1,587,198 for extrapolation. The study included 51,562 beds, of which 40·3% were in the medical specialty followed by surgery (30%), obstetrics and gynecology (17·1%) and pediatrics (12·6%). Each of these four categories included general, specialties and sub-specialties beds. As demand for blood can occur only in occupied beds in a health care facility, we determined the bed occupancy rate (BOR) adjusted clinical demand per bed as well. After estimating the total demand in a health care facility, we adjusted for the actual bed occupancy. For this calculation, we used the total number of beds, bed occupancy rate to calculate the occupied beds and the total clinical demand in the health care facilities. Bootstrap method was used to get narrow confidence intervals for the estimation when the number of facilities per stratum was less than 50.

After estimating the total demand in a health care facility, it was adjusted for the actual bed occupancy. For this, we used the total number of beds, bed occupancy rate and the total clinical demand in the health care facilities.

## Estimated number of the eligible donor population

To estimate the eligible donor population in the country, we considered the inclusion and exclusion criteria defined by the National Blood Transfusion Council (NBTC) and Drug and Cosmetics Act of India (1940), as amended up to 31st December 2016 [12]. We included the adult population between the age group of 18 to 65 years, which is 58·6% (745 million) of the total population in 2017 [13]. We excluded the pregnant women (estimated at 27 million) and people with anaemia, (adult male, 22·7% and adult female, 53%), hypertension (25·3%), diabetes (7·5%) and co-existence of any two or all the three conditions [7, 14]. We did not exclude the estimated 2·5 million cancer patients as anaemia coexists in the majority of cancer patients [15–17] Besides, we factored 5% for other temporary deferrals. Applying all these criteria, we conservatively estimated 402 million as eligible donor population in India.

## Ethical approval

The study protocol was reviewed and approved by the Institutional Review Board (IRB) of Christian Medical College and Hospital, Vellore, India. We obtained approval from Technical Resource Group (TRG), Research and Development, National AIDS Control Organization (NACO), Ministry of Health and Family Welfare, Government of India and Associate Director for Science (ADS) and Associate Administrator for Science (ATSDR), the Centers for Disease Control and Prevention (CDC), Atlanta, USA.

For data collection from health care facilities, the National AIDS Control Organization (NACO), Ministry of Health and Family Welfare, Government of India, sent a letter of invitation to the medical head/administrator of all selected health care facilities to participate in the study. Following this, the study team approached the heads of the institutions over the phone to obtain their oral consent for the participation and convenient time for data collection. Before the data collection in the facilities, the study investigator obtained informed written consent from the heads of the institutions. For the Delphi exercises, we sent a letter of invite requesting the health care providers (technical experts) to participate in the study and their agreement to participate by a response mail was considered as their consent.

## Results

The characteristics of healthcare facilities are mentioned in Table 1. The majority (76·5%) were in urban, around 57% were owned by the private sector, and two-thirds (67%) of the private sector were for-profit facilities. The average bed occupancy rates for primary, secondary and tertiary care facilities were 60·4% (SD: 25.9), 72·9% (SD: 24·7) and 83.9% (SD: 17·3) respectively. A third of the facilities (34%) had an attached blood bank.

### Estimated national clinical demand

According to the study, the total clinical demand of 251 health facilities with 51,562 beds was 474,627 whole blood units. Based on this, the total clinical demand was estimated at 14·61 million units (95% CI: 14·59–14·62) of whole blood, which is equivalent to a requirement of 36·3 donations per 1,000 eligible persons, considering the existing blood transfusion practices and

**Table 1. Details of healthcare facilities.**

| Components | Total | North Region | East Region | West Region | South Region | Northeast Region |
|---|---|---|---|---|---|---|
| **Location** | (n = 251) % | (n = 52) % | (n = 48) % | (n = 48) % | (n = 49) % | (n = 54) % |
| *Rural* | 59 (23.5) | 10 (19.2) | 12 (25.0) | 13 (27.1) | 14 (28.6) | 10 (18.5) |
| *Urban* | 192 (76.5) | 42 (80.8) | 36 (75.0) | 35 (72.9) | 35 (71.4) | 44 (81.5) |
| **Ownership** | | | | | | |
| *Private* | 144 (57.4) | 28 (53.8) | 28 (58.3) | 29 (60.4) | 28 (57.1) | 31 (57.4) |
| *Public* | 107 (42.6) | 24 (46.2) | 20 (41.7) | 19 (39.6) | 21 (42.9) | 23 (42.6) |
| **Level of care** | | | | | | |
| *Primary* | 107 (42.6) | 23 (44.2) | 21 (43.8) | 17 (35.4) | 24 (49.0) | 22 (40.7) |
| *Secondary* | 113 (45.0) | 23 (44.2) | 21 (43.8) | 24 (50.0) | 19 (38.8) | 26 (48.1) |
| *Tertiary* | 31 (12.4) | 6 (11.5) | 6 (12.5) | 7 (14.6) | 6 (12.2) | 6 (11.1) |
| **Type of facilities (Public)** | | | | | | |
| *CHC* | 45 (42.1) | 10 (41.7) | 9 (45.0) | 5 (26.3) | 10 (47.6) | 11 (47.8) |
| *SDH/DH* | 47 (43.9) | 11 (44.9) | 8 (40.0) | 11 (57.9) | 8 (38.1) | 9 (39.1) |
| *MC* | 15 (14.0) | 3 (12.5) | 3 (15.0) | 3 (15.8) | 3 (14.3) | 3 (13.0) |
| **Type of facilities (Private)** | | | | | | |
| *Profit* | 96 (66.7) | 14 (50.0) | 23 (82.1) | 18 (62.1) | 20 (71.4) | 21 (67.7) |
| *Not for profit* | 48 (33.3) | 14 (50.0) | 5 (17.9) | 11 (37.9) | 8 (28.6) | 10 (32.3) |
| **Total number of beds** | | | | | | |
| *< = 50* | 111 (44.2) | 23 (44.2) | 22 (45.8) | 20 (41.7) | 24 (49.0) | 22 (40.7) |
| *51 Beds to 200* | 84 (33.5) | 18 (34.6) | 14 (29.2) | 15 (31.3) | 13 (26.5) | 24 (44.4) |
| *More than 200* | 56 (22.3) | 11 (21.2) | 12 (25.0) | 13 (27.1) | 12 (24.5) | 8 (14.8) |
| **Bed occupancy rate** | | | | | | |
| *Less than 25%* | 15 (6.0) | 3 (5.8) | 4 (8.3) | 0 | 7 (14.3) | 1 (1.9) |
| *25% to 75%* | 132 (52.6) | 29 (55.8) | 23 (47.9) | 27 (56.3) | 22 (44.9) | 31 (57.4) |
| *More than 75%* | 104 (41.4) | 20 (38.5) | 21 (43.8) | 21 (43.8) | 20 (40.8) | 22 (40.7) |
| **Attached blood bank** | | | | | | |
| *Yes* | 86 (34.3) | 20 (38.5) | 14 (29.2) | 16 (33.3) | 18 (36.7) | 18 (33.3) |
| *No* | 165 (65.7) | 32 (61.5) | 34 (70.8) | 32 (66.7) | 31 (63.3) | 36 (66.7) |

* Percentage in parenthesis.

the amount of component separation in the facilities in India [18]. The estimated demand for medical specialty was 6·0 million units (41·2%), followed by surgery 4·1 million (27·9%), obstetrics and gynecology 3·3 million (22·4%) and pediatrics 1·2 million (8·5%) units (Table 2).

**Table 2. Estimated national clinical demand.**

| | Study data n-251 | | | |
|---|---|---|---|---|
| Specialty | Number of beds | Clinical demand in whole blood units | Beds | Estimated national demand in whole blood units* (95% CI) |
| **Medicine** | 20,779 | 195,434 | 639,641 | **6,015,910** (6,012,223–6,019, 597) |
| **Surgery** | 15,469 | 132,370 | 476,159 | **4,074,654** (4,071,294–4,078,014) |
| **Obstetrics & Gynaecology** | 8,817 | 106,312 | 271,411 | **3,272,529** (3,269,405–3,275,652) |
| **Pediatrics** | 6,497 | 40,511 | 199,987 | **1,247,022** (1,244,929–1,249,115) |
| **Total** | **51,562** | **474,627** | **1,587,198** | **14,610,116** (14,597,852–14,622,378) |

*Estimated the national demand by extrapolating the study data (demand and beds) to the total number of estimated beds in the country.

**Table 3. BOR adjusted demand per bed in whole blood units.**

| Description | Demand per bed | BOR adjusted demand per bed | 95% CI* | |
|---|---|---|---|---|
| | | | Lower | Upper |
| **Region** | | | | |
| North | 8.2 | 10.3 | 7.9 | 12.5 |
| East | 9.3 | 11.3 | 10.5 | 15.9 |
| West | 9.7 | 12.1 | 9.5 | 14.5 |
| South | 8.6 | 10.7 | 9.9 | 15.8 |
| North East | 10.5 | 11.7 | 9.9 | 15.2 |
| **Location** | | | | |
| Rural | 6.5 | 9.1 | 6.4 | 11.8 |
| Urban | 9.5 | 11.4 | 11.2 | 14.0 |
| **Level of care** | | | | |
| Primary | 4.6 | 7.6 | 7.3 | 9.6 |
| Secondary | 9.4 | 12.3 | 10.9 | 13.8 |
| Tertiary | 9.6 | 10.9 | 8.8 | 12.4 |
| **Ownership** | | | | |
| Public | 9.5 | 10.3 | 7.7 | 10.6 |
| Private | 8.9 | 12.7 | 12.0 | 15.3 |
| **Availability of BBs** | | | | |
| With blood bank | 9.7 | 11.3 | 9.9 | 13.5 |
| Without blood bank | 6.9 | 10.5 | 7.5 | 11.6 |
| **Total** | 9.2 | **11.2** | **10.7** | **13.2** |

95% CI is based on bootstrap method using institution-level BOR adjusted demand per bed.

## Clinical demand per bed

The crude clinical demand per bed was 9·2 units (95% CI 7·5–9·5) and bed occupancy rate (BOR) adjusted clinical demand per bed was 11·2 (95% CI: 10·7–13·2) units per annum. It was found to be relatively lower in the northern region, rural areas, primary care facilities, public facilities, and in facilities without an attached blood bank (Table 3).

## Major contributors of clinical demand

The diseases or conditions that recorded more than one percent of total specialty demand, medical, surgical, O&G and pediatrics are mentioned in Table 4.

**Medical specialty.** Most of the demand was for nutritional anaemia (32·8%) which is equivalent to 1·97 million units followed by end-stage renal diseases (ESRD) (9·7%), gastrointestinal bleed (5·8%), chronic liver disorder (5·2%) and leukemia (5%). The demand for communicable diseases such as dengue and malaria were 0·45 million units, which is 7·5% of the medical demand.

**Surgical specialty.** The estimated demand for orthopedic surgeries was 25%, which is equivalent to around one million units. The demand for polytrauma and road traffic accidents was 0·46 million units which is around 11% of surgical demand. The demand for oncology surgery was 0·34 million units which is 8·4% of surgical demand.

**Obstetrics & Gynecology specialty.** A greater proportion of demand (34·2%) was for anaemia in pregnancy, which amounts to 1·1 million units for an estimated 27·7 million pregnant women in a year. Maternal and pregnancy-related complications such as ectopic pregnancy, antepartum hemorrhage–abruptio/placenta praevia, placenta accrete, and

**Table 4. Major contributors of demand—medicine, surgery, obs & gyn and paediatrics (whole blood units).**

| Medicine | Demand | % | Surgery | Demand | % |
|---|---|---|---|---|---|
| Nutritional Anaemia | 1,974,517 | 32.8 | Orthopaedic Surgeries | 1,015,636 | 24.9 |
| End-stage renal disease (ESRD) | 586,509 | 9.7 | Abdominal surgeries | 555,298 | 13.6 |
| Gastro-Intestinal Bleed | 346,463 | 5.8 | Trauma | 403,361 | 9.9 |
| Dengue | 334,027 | 5.6 | Oncology surgeries | 340,844 | 8.4 |
| Chronic Liver Disorder | 314,572 | 5.2 | Head and Neck surgeries | 275,802 | 6.8 |
| Leukaemia | 301,982 | 5 | Adult cardiac surgeries | 200,818 | 4.9 |
| Oncological conditions | 221,515 | 3.7 | General surgeries | 182,534 | 4.5 |
| Others | 199,751 | 3.3 | Neurosurgical Procedures | 154,031 | 3.8 |
| Idiopathic Thrombocytopenic Purpura | 169,984 | 2.8 | Burns | 141,718 | 3.5 |
| Chronic Kidney Disease | 141,202 | 2.3 | Urology | 100,409 | 2.5 |
| Hemolytic anaemias | 119,531 | 2 | Thoracic surgeries | 84,988 | 2.1 |
| Malaria | 113,528 | 1.9 | Others | 63,287 | 1.6 |
| Myeloma & Lymphoma | 100,199 | 1.7 | Coronary artery bypass graft (CABG) | 57,315 | 1.4 |
| Autoimmune hemolytic anaemias | 94,689 | 1.6 | Road Traffic Accident (RTA) | 52,852 | 1.3 |
| Haemophilia | 61,843 | 1 | Pediatric cardiac surgeries | 41,709 | 1 |
| Rheumatic Heart Disease | 60,242 | 1 | Fracture | 39,031 | 1 |
| Disseminated Intravascular Coagulation | 59,627 | 1 | | | |
| Sepsis | 58,826 | 1 | | | |
| Others | 756,903 | 12.6 | Others | 365,021 | 8.8 |
| **Sub-total** | **6,015,910** | **100** | **Sub-total** | **4,074,654** | **100** |
| **Obstetrics & Gynaecology** | **Demand** | **%** | **Pediatrics** | **Demand** | **%** |
| Anaemia in pregnancy | 1,120,428 | 34.2 | Hemolytic anaemias–Thalassemia | 294,363 | 23.6 |
| Postpartum hemorrhage (PPH)–Caesarean, atonic, ret. Placenta, Traumatic Post-Partum Haemorrhage | 484,119 | 14.8 | Severe Nutritional Anaemia | 175,885 | 14.1 |
| Abnormal Uterine Bleeding | 190,145 | 5.8 | Leukemia | 102,164 | 8.2 |
| LSCS | 153,052 | 4.7 | Others | 85,418 | 6.8 |
| Abortions | 148,496 | 4.5 | Sepsis & Disseminated intravascular coagulation (DIC) | 57,746 | 4.6 |
| Hysterectomy–Abnormal uterine bleeding (AUB)AUB/ Prolapse/ Adenomysosis/ Endometriosis-Fibroids/ PID | 123,654 | 3.8 | Dengue | 54,545 | 4.4 |
| APH–Placenta praevia | 96,104 | 2.9 | Neonatal transfusions–Low Birth Weight | 52,575 | 4.2 |
| Ectopic pregnancies | 89,516 | 2.7 | Neonatal transfusions–Sepsis | 44,325 | 3.6 |
| Antenatal care (ANC) | 89,177 | 2.7 | Malaria | 37,184 | 3 |
| Fibroids–Myomectomy | 88,900 | 2.7 | Aplastic Anaemia | 33,860 | 2.7 |
| Placenta praevia—accrete/accreta | 81,543 | 2.5 | Gastrointestinal (GI) bleeding | 31,890 | 2.6 |
| Antepartum haemorrhage (APH)–Abruptio | 74,894 | 2.3 | Chronic renal disease | 29,396 | 2.4 |
| Carcinoma ovary | 71,724 | 2.2 | Neonatal transfusions–birth asphyxia/ trauma | 26,472 | 2.1 |
| Cancer cervix | 62,150 | 1.9 | Neonatal Jaundice | 24,256 | 1.9 |
| Postnatal care (PNC) with Anaemia | 58,826 | 1.8 | Thrombocytopenia | 20,870 | 1.7 |
| Hematology–Factor deficiency, All thrombocytopenia, Anticoagulation, | 43,896 | 1.3 | Haemophilia–severe | 19,208 | 1.5 |
| | | | Sickle Cell Anaemia | 17,792 | 1.4 |
| | | | Congenital Heart Disease | 12,713 | 1 |
| Others | 2,95,905 | 9.2 | Others | 126,360 | 11.2 |
| **Sub-total** | **3,272,529** | **100** | **Sub-total** | **1,247,022** | **100** |

postpartum hemorrhage contributed around 25·2% which is around 0·8 million units. Whereas, gynecological cancers, accounted for around 4·1% (0·13 million units) of the gynecology demand.

**Table 5. Current demand, supply and utilization of whole blood and components (units).**

| Specialty | Whole Blood | Red cells | Plasma | Platelets | Cryo precipitate |
|---|---|---|---|---|---|
| **Estimated national demand (in 'thousands; 95% CI)** | 6,354 (6,350–6,358) | 6,613 (6,609–6,617) | 2,208 (2,206–2,211) | 1,862 (1,859–1,864) | 221 (220–222) |
| **Total supply (in 'thousands)** | 5,882 (5.879–5,886) | 6,065 (6,062–6,069) | 1,900 (1,898–1,903) | 1,442 (1,440–1,445) | 203 (203–204) |
| **Total utilization (in 'thousands)** | 5,807 (5,803–5,812) | 6,016 (6,012–6,020) | 1,838 (1,836–1,841) | 1,407 (1,405–1,410) | 192 (192–193) |

**Pediatric specialty.** Hemolytic anaemia (predominantly thalassemia) is the leading contributor to the demand for pediatrics specialty (23·6%), followed by severe nutritional anaemia (14%) and leukemia (8·2%). Dengue and malaria contribute to 7·4% of clinical demand, which amounts to 91,729 units. Neonatal conditions such as very low birth weight (VLBW), sepsis, birth asphyxia/trauma and neonatal jaundice together account for 11·8% amounting to 0·15 million units.

**National demand for blood components.** The whole blood demand was estimated at 6·35 million units (95% CI: 6·34–6·36), red cell concentrates 6·61 million units (95% CI: 6·60–6·62), plasma 2·21 million units (95% CI: 2·20–2·22), platelets- 2·0 million units (95% CI: 1·99–2.02) and 0·22 million units (95% CI: 0·21–0·22) cryoprecipitate.

**Current supply and utilization of whole blood and components.** The supply against demand was estimated at 92·6% for whole blood, 91·7% for red cells, 86% for plasma, 77·5% for platelets, and 92·1% for cryoprecipitate. Similarly, the utilization against supply was reported at 98·7% for whole blood, 99·2%, 96·8% 97·6% and 94·6% for red cells, plasma, platelets and cryoprecipitate respectively (Table 5).

## Discussion

This national-level cross-sectional study is the first of its kind in India to estimate the gaps between clinical demand, supply, utilization. The estimated national clinical demand of 36·3 donations per 1,000 eligible population, is closer to the WHO's suggested donation rate of one to three percent of a country's population which would be sufficient to address the requirement [19].

The clinical demand primarily depends on the morbidity pattern, access to health services and capacity of health systems for service provision which vary between countries and within countries. Moreover, there has been a changing trend in the demand for blood over a period of time. A study in Tanzania estimated a demand of 6·2 blood donations per 1,000 persons after adjusting for irrational use and assuming 100% component separation which is significantly lower compared to our study [20]. In developed countries, the advent of modern technology, less invasive surgical procedures, pharmacological alternatives to transfusion, changing approach to the treatment of oncological conditions and effective patient blood management strategies have led to a reduction in demand and usage of blood. For instance, in the United States, there has been a decreasing trend of blood collection and transfusion that has fallen 25% since 2008 [21]. In Switzerland, a decrease of 18·6% of red cells demand has been predicted between 2013 and 2035 [22]. Germany indicated a 13.5% decrease in in-hospital transfusion demand from 2005 to 2015 [23].

In contrast, increasing demand is projected in underdeveloped and developing countries with improving access to health care. The Lancet Global Surgery Commission reports that there are about 143 million unmet surgical procedures in Low and Middle-income countries (LMIC) with a target to work towards achieving 5,000 procedures per 100,000 population, which may, in fact, increase the demand for blood [24].

The clinical demand per occupied bed is one of the critical indicators for planning transfusion services as demand is generated only from occupied beds in health care facilities. The demand per bed was higher in private facilities which could be due to the provision of highly specialized services such as dialysis and specialized surgeries that demand a higher volume of blood transfusions. Besides, the relative ease of availability of blood to the private health facilities, affordability of patients and possible non-adherence to strict guidelines could be the other contributing factors. The relatively higher BOR adjusted demand per bed in secondary care facilities could be due to the proximity, easy access, availability of all specialty services, and rational use of blood in tertiary care facilities which are mostly academic institutions.

## Distribution of demand

According to our study, the demand for medical specialty was the highest, followed by surgery, obstetrics & gynecology and pediatrics respectively. Unlike the pattern in India, a study in Tanzania estimated a demand of 32·6% for medical specialty, 11.8% for surgery, 28·4% for obstetrics and gynecology and 26·8% for paediatrics [20].

In the medical specialty, nutritional anaemia (32%) which is an avoidable public health concern was the major contributor for the demand for blood followed by end-stage renal disease. Though theoretically, less than 7 g/dl require transfusion in anaemia, the actual clinical practice is not known clearly. In patients with co-existing morbidities that compromise cardiac function, and in places where critical care services are not available, the threshold may be higher. However, the study provides crucial evidence for the urgent need for anaemia screening and effective nutrition programs to prevent avoidable transfusion due to severe anaemia. The concerns linked to the increasing prevalence of non-communicable diseases in India are beginning to manifest with increasing numbers of patients with ESRD; many of these patients will experience anaemia and will require ongoing dialysis or renal transplantation, both of which are linked to increased demand for blood. Though early parenteral iron supplementations with erythropoietin (EPO) were found to be beneficial for the management of anaemia, especially in end-stage renal disease patients, it is not widely practiced as it is not feasible and affordable in India [25].

Among the seasonal febrile illnesses, dengue continues to be on the forefront, although experts agree that many of the platelet transfusions are unjustified, more often driven by caution and pressure from family members [26].

Around 28% of the total demand is for surgical specialties. The Lancet Commission on Global Surgery recommends at least 15 units per 1,000 persons per year which would be a significant volume of blood in India [24]. Besides, the increasing number of road traffic accidents implies that the demand for blood for trauma surgery will continue to be a significant contributor [27]. Increasing access to early diagnosis and treatment for oncological surgeries have also led to increased demand for blood which reflected a relatively higher demand of 0·34 million units [28–30].

In obstetrics and gynecology, the higher demand for anaemia during pregnancy, post-partum bleeding and abnormal uterine bleeding stress the importance of ensuring timely availability of blood to prevent avoidable maternal deaths. The higher demand for blood for caesarean section delivery (4·7%) reflects the increasing trend of surgical deliveries in the country. In pediatrics, the continuing burden of inherited hemolytic anaemia including thalassemia is one of the reasons for the high proportion of demand (23·6%) among children. Despite the substantial investment and efforts through national programs, the blood demand for severe nutritional anaemia among children indicates the need for revisiting the current strategies and programs related to anaemia.

### Demand for blood components

The study provided evidence related to the specific demands for whole blood and components as well. It highlights the significant demand for whole blood in the country despite the continuous emphasis on rational use of blood through appropriate components. The possible reasons for the higher demand for whole blood could be the lack of knowledge and awareness among clinicians regarding component therapy, archaic clinical practices, lack of availability of blood component separation units, and sub-optimal component separation. Though the reported component separation rate was reported to be better (71%), the other reasons for the low demand for components need to be analyzed and addressed [18]. The demand for cryoprecipitate was found to be the lowest as it is used specifically for replacement of coagulation factor FVIII: C, von Willebrand antigen and fibrinogen in inherited and acquired conditions which are uncommon. Moreover, purified plasma-derived or recombinant clotting factor concentrates are easily accessible through national programs, which is an alternative for cryoprecipitate.

### Supply against demand

Taking into consideration of the current supply against the demand for whole blood (92·6%) and red cells (91·7%), if the current supply in terms of whole blood donation is assumed to be around 93%, it will be equivalent to 33·8 donations against the estimated clinical demand of 36·3 donations per 1,000 eligible persons in India. The demand and supply gap of 2·5 donations per 1,000 persons, could be due to low voluntary blood donation, sub-optimal component production, injudicious demand and archaic practices. In 2013, the World Health Organization (WHO) reported significant variations in blood donation rates which were 32·1, 14·9, 7·8 and 4·6 per 1,000 persons per year in high, upper-middle, lower-middle and lower-income countries respectively. In the South-East Asia WHO region, the donation rates ranged from 1·8 to 30·8 (median 7·9) [31]. In 2006, the Sub-Saharan African countries collected less than half of their estimated minimum blood donation requirement [32] and Tanzania reported an unmet demand of 2·5% due to non-availability of blood [20]. There is a documented variation in the donation rates between regions and countries. However, it needs to be carefully interpreted as the computation of the eligible population who can donate may be different.

### Utilization against supply

Though the reported utilization rate seems to be reasonable, the influence of supply-induced demand, irrational usage of blood due to societal pressure, defensive practice and other possible factors need to be studied further. The relatively lower utilization of plasma, platelets and cryoprecipitate could be due to the shorter shelf life and lack of temperature-controlled transportation system that could result in degradation of product [33]. This highlights the need for rational patient blood management practices and capacity building of clinicians and blood transfusion personnel in the country.

### Limitations

We assumed that the weights will not have much impact on the estimates, since we had a high precision in the sample size estimation to account for the clustering effect. The study has limitations because it did not focus on analyzing clinician's requests for each patient to determine the rationality of demand. Our study did not evaluate clinician ordering practices to determine if they were appropriate. The indications for blood transfusions and adherence to guidelines

could vary between clinicians, health facilities and regions which could be potential limitations of the study.

The facility-wise supply and utilization data for each component were obtained as a percentage. Moreover, the declared official number of hospital beds may not be accurate in the facilities. The study did not factor in the possible disasters and epidemics that would generate demand which depends on their nature and severity.

## Conclusion

Our study estimated a national demand of 14·6 million whole blood units, which translates to 36·3 donations per 1,000 eligible persons per year. The estimated BOR adjusted demand per bed was at 11·2 units per year. Although the gap between supply and demand is only 2·5 donations per 1,000 persons, which is around one million units, it requires sustained and concerted efforts from all stakeholders to address the gap.

The demand and supply gap emphasizes the need for increasing the awareness about repeat VNRBD; optimizing the availability of blood components through efficient blood component separation units; promoting modern principles of patient blood management and strengthening capacities of human resources in the blood transfusion system, to ensure universal access to blood and components in India.

## Author Contributions

**Conceptualization:** Joy John Mammen, Edwin Sam Asirvatham, Jeyaseelan Lakshmanan, Arvind Pandey, Bimal Charles, Sunil D Khaparde, Sunita Upadhyaya, Shobini Rajan.

**Data curation:** Charishma Jones Sarman, Varsha Ranjan.

**Formal analysis:** Joy John Mammen, Edwin Sam Asirvatham, Jeyaseelan Lakshmanan, Charishma Jones Sarman, Varsha Ranjan, Thenmozhi Mani.

**Investigation:** Charishma Jones Sarman, Varsha Ranjan.

**Methodology:** Joy John Mammen, Edwin Sam Asirvatham, Jeyaseelan Lakshmanan, Charishma Jones Sarman, Arvind Pandey, Varsha Ranjan, Bimal Charles, Thenmozhi Mani, Sunita Upadhyaya.

**Project administration:** Bimal Charles, Shobini Rajan.

**Supervision:** Joy John Mammen, Edwin Sam Asirvatham, Charishma Jones Sarman, Arvind Pandey, Varsha Ranjan, Bimal Charles, Sunil D Khaparde, Sunita Upadhyaya, Shobini Rajan.

**Validation:** Joy John Mammen, Edwin Sam Asirvatham, Charishma Jones Sarman, Varsha Ranjan, Thenmozhi Mani.

**Writing – original draft:** Joy John Mammen, Edwin Sam Asirvatham, Jeyaseelan Lakshmanan, Charishma Jones Sarman, Varsha Ranjan, Sunita Upadhyaya, Shobini Rajan.

**Writing – review & editing:** Joy John Mammen, Edwin Sam Asirvatham, Jeyaseelan Lakshmanan, Charishma Jones Sarman, Arvind Pandey, Bimal Charles, Thenmozhi Mani, Sunil D Khaparde, Sunita Upadhyaya, Shobini Rajan.

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
