## [Decision Letter · Decision Letter 0]

13 Sep 2021

PONE-D-21-22354Demand and supply dichotomy of blood and components: A National level estimation study in IndiaPLOS ONE

Dear Dr. Mammen,

Thank you for submitting your manuscript to PLOS ONE. After careful consideration, we feel that it has merit but does not fully meet PLOS ONE’s publication criteria as it currently stands. Therefore, we invite you to submit a revised version of the manuscript that addresses the points raised during the review process. The manuscript presents interesting data and is of great interest given the scale of the analysis. However, there are several important limitations that preclude to accept the manuscript in its current form. We invite you to submit a revised version of the manuscript that addresses the points below:

Please keep special attention to the very useful Reviewers' suggestions, including:

1. Need to revision of the manuscript and rephrasing for clarity.

2. Preparation of a flow chart to describe the sampling strategy. Please describe not only general logic of sampling, but also provide the exact numbers of different facilities (private, public, tertiary, etc) considered during the sampling, and the exact numbers of finally selected facilities.

3. Need to a clear distinction between blood and components.

4. More details on how is the diagnosis and transfusion related information have been documented.

Additionally, I suggest to:

1. Please provide the reference to the the National Health Profile (2017).

2. The demand is one of the core concepts of the manuscript, and much more details on its calculation should be presented. The phrase "We computed the demand for whole blood or components by extrapolating the sum of whole blood or components demanded by each institution." is not clear. Please provide more details also on statistical methods (first of all about BOR adjustment).

3. Please provide in the footnote of all tables the definitions of abbreviations used.

4. In the Table 2, please use the same digit capacity for the "Clinical demand in whole blood units" for the columns related to study data and national estimates.

5. In the Table 3, please introduce a column with "demand per bed before adjustment" data.

6. The number of blood transfusions for the Nutritional Anemia is impressive. Could you please comment in the Discussion which haemoglobin level is the indication of blood transfusion for this population?

7. The second most prevalent category requiring blood transfusion is patients with End-stage renal disease. Could you please comment in the Discussion about availability of erythropoetin and parenteral iron medications in India, as well as which haemoglobin level is the indication of blood transfusion in this particular disease? You provided separate estimates for "End-stage renal disease" and "Chronic Kidney Disease" - could you please comment on it?

8. In the Table 3 the "Sepsis" and "Disseminated Intravascular Coagulation" are 2 separate categories for adults, but one merged category for pediatric care. Please could you explain why the same questionnaires applied have led to different categories?

9. In the tables, please use "in thousands" instead of "(in ‘000)".

10. It would be better to use "general medicine" instead of "medical specialty", where appropriate..

11. While only 30-40% of considered facilities have their own blood bank, could you please describe some details on how the blood or components are distributing from facilities with to those without blood bank? It would be also interesting to know more about blood donation in India - whether donors have some incentives, are there any cultural or religious patterns of blood donation, etc.

12. Please indicate the volume for the whole blood and components used. Even if their volume per unit is standard, the clear definition is always a good choice.

13. Please avoid abbreviations in the references where possible. Please control for the references completeness, for example "IIPS. National Family Health Survey-2017 (NFHS-4) New Delhi: International Institute for Population Sciences, Ministry of Health and Family Welfare, 2017" would be better to supply with ISBN or web-link.

14. It is up to your choice, but it would be useful to share the questionnaire in the supplement, to give the opportunity to reproduce the study for other LMICs research groups.

We look forward to receiving your revised manuscript.

Kind regards,

Boris Bikbov

Academic Editor

PLOS ONE

Journal Requirements:

2.Please note that Consent info is not provided, please provide your consent info. 

Reviewers' comments:

Reviewer's Responses to Questions

**Comments to the Author**

1. Is the manuscript technically sound, and do the data support the conclusions?

Reviewer #1: Yes

Reviewer #2: Partly

Reviewer #3: Partly

2. Has the statistical analysis been performed appropriately and rigorously? 

Reviewer #1: Yes

Reviewer #2: Yes

Reviewer #3: Yes

3. Have the authors made all data underlying the findings in their manuscript fully available?

Reviewer #1: No

Reviewer #2: Yes

Reviewer #3: Yes

4. Is the manuscript presented in an intelligible fashion and written in standard English?

Reviewer #1: Yes

Reviewer #2: No

Reviewer #3: Yes

5. Review Comments to the Author

Reviewer #1: It’s my opinion that the manuscript is fairly good and well written. The information supplied is also relevant and timely. I have a few questions and some minor concerns. First, the authors should consider the use of a flow chart to describe the sampling strategy. The current approach appears to be long-winded, hence difficult to follow. Further, the authors should have incorporated more references on blood demand from other jurisdictions e.g. regional studies within India or Asia. Although these studies are rare, I’m sure a sufficient number or papers can be found in existing digital repositories. In addition, reference 17 and 22 are similar.

Reviewer #2: An important topic chosen by a group of authors from different fields, kudos to them. I have few observations which I feel you will be able to answer and further improve the article.

1. Introduction

a) Para 1 last line needs rephrasing

b) para 2 what programmes are being talked about

c) para 3 there is repetition and explanation of the same words, avoid semantics and reduce unnecessary words

2.The aim of the study in the abstract and main body is different, please explain.

3. The study used data from various different institutes, was the ethic clearance from each participating institute taken.

4. Table one says beds <30, 31-200 etc while text in methods says beds <50 as primary in pvt sector, please justify.

5. Table 4 needs to be made more reader friendly by reducing unnecessary rows.

6. Please explain 'Discussion " para 2 initial 4 lines

7. Discussion

a) Demand for blood components - has been written and appears out of context. a comment about the clinician's know;edge regarding components was not surveyed and should not be commented upon.

b) The discussion needs to be more specific and needs rewriting all over.

8. Please remove all the unnecessary comments from the conclusion which were not part of the aim & objectives. Please stick to the study per se and not the general comments

Thanks

Reviewer #3: Reviewer´s comments

Ms. No. PONE-D-21-22354

Ms. Title Demand and supply dichotomy of blood and components: A national level estimation study in India

Authors: Mamman J et al

The aim of this study, with authors representing health authorities in India, was to estimate clinical demand, supply and utilization of (whole?) blood and blood components, based on data from randomly selected facilities and then by extrapolating the study data, give estimates for the whole country.

General comments

The aim to estimate the supply and need of blood is ambitious and important, and it is a challenge, especially in a country like India, with a very large population and different levels of health care.

The effort and work are impressive, and it is an interesting descriptive report of the blood supply and usage, but the final estimations are based on many uncertain assumptions. It is not convincingly shown that the selected facilities are representative, the indications for blood transfusions varies and are in many parts not appropriate, according to guidelines in developed countries. The authors make reservations in the discussion and discuss the limitations of the study, but this could be further emphasized.

Specific comments

Title

I do not fully understand the title: Blood and components? The availability and use of blood components vs whole blood and the demand of different blood components are not analysed.

Something like “The blood supply in India-An estimation of available donors and demand and usage of blood transfusions on a national level”………?

Abstract

line 31 blood and components? Do you mean whole blood and blood components?

Line 35-36 are the regions, states and facilities all randomly selected? Not chosen as representative for different parts of the country and health care level?

Line 42-44 The study data, from where the national estimations are calculated should be presented.

Line 50-53 The conclusion should be more specific. What is the main message? Some of the findings discussed on page 15, line 282-306 and page 19 line 390-395?

Introduction

The introduction is informative, describing 2760 blood banks, public or private, often attached to hospitals, component preparation of 53% of the collected blood.

To interpret the data and results a short paragraph is requested of costs in the health care; is the health care free or paid for in public facilities? In private facilities? Are blood transfusions free or paid for? Are blood donors non-remunerated or compensated? Are relatives asked to donate?

Methods

Line 92-98. It would be nice with a map of India where you marked the regions and states and the selected facilities.

Line 98-100 how was the facilities randomly selected?

Line 98-99 define primary, secondary and tertiary health care facilities, with respect to blood transfusions/blood donations? It is partly explained later in line 108, “based on expert´s consensus”? Do you have a reference, documentation or is it an assumption?

Data collection

Line 133-134 How is the diagnosis and transfusion related information documented? Is it similarly documented in the different facilities?

Line 136-138 Please describe how you retrieved the data from the systems? Electronically or manually? Do all centers have electronic hospital and blood bank records?

Line 138 How was the data validated?

Data analysis and computation of clinical demand

Line 140 How was data entered in SPSS?

Line 157-158 Are the indications for whole blood and components different or is it a matter of tradition and availability?

Results

Line 219 Is nutritional anemia an appropriate indication for blood transfusion?

Discussion

The discussion includes many interesting aspects. Will the need of blood transfusions increase or decrease in India, referring to international trends? Are blood transfusions used on appropriate indications or is education primary needed?

6. PLOS authors have the option to publish the peer review history of their article (what does this mean?). If published, this will include your full peer review and any attached files.

Reviewer #1: No

Reviewer #2: **Yes: **DR SUSHIL CHAWLA

Reviewer #3: No

---

## [Author Response · Author response to Decision Letter 0]

21 Feb 2022

A separate file indicating the reviewers comments and responses has been attached. 

PONE-D-21-22354

Demand and supply dichotomy of blood and components: A National level estimation study in India

Response: The title has been changed as advised by the reviewers as below

The Clinical Demand and Supply Of Blood in India: A National Level Estimation Study 

Editors Comments

1. Need to revision of the manuscript and rephrasing for clarity.

We have substantially revised and rephrased the manuscript for clarity

2. Preparation of a flow chart to describe the sampling strategy. Please describe not only general logic of sampling, but also provide the exact numbers of different facilities (private, public, tertiary, etc) considered during the sampling, and the exact numbers of finally selected facilities.

We have provided a flow with the number of facilities included in the study. 

During the study, we were able to obtain the actual number of public health care facilities. (4,833 Community Health Centres (CHCs), 987 sub-divisional hospitals, 722 district hospitals and 381 medical colleges in the country). However, there was no single list private health care facilities in India. Therefore, we collected the state-wise list of private health care facilities from multiple sources such as Indian Medical Association of India, list of public/private hospitals empanelled under Rashtriya Swasthya Bima Yojana (RSBY), list of private hospitals empanelled under Central Government Health Scheme (CGHS) and other reliable sources. We triangulated the data from all sources to arrive at the tentative list of private facilities and obtained the samples from the list.

We have provided a flow chart for describing the sampling strategy. 

3. Need to a clear distinction between blood and components.

In our study, we included both whole blood and its components. Despite the significant increase in the use of components, whole blood is still being requested and transfused in a significant proportion of health facilities in the country. Therefore, in our analysis, each unit of whole blood demand is equated to a single whole blood donation. Similarly, each unit of component demand is equated to a single whole blood donation. To avoid overestimation, we considered the component that recorded the highest demand among all components. It is due to the fact that one unit of whole blood can provide a minimum of one unit of components (Red cells, Plasma, platelets and cryo) each.

We have made the necessary changes in the manuscript for clarity.

4. More details on how is the diagnosis and transfusion related information have been documented.

The diagnosis and the transfusion-related information have been obtained from the Hospital records retrospectively for a period of 6 months. The details are provided in the data collection section of the manuscript.

Additionally, I suggest to:

1. Please provide the reference to the National Health Profile (2017).

Yes, we have provided the reference now.

2. The demand is one of the core concepts of the manuscript, and much more details on its calculation should be presented. The phrase "We computed the demand for whole blood or components by extrapolating the sum of whole blood or components demanded by each institution." is not clear. Please provide more details also on statistical methods (first of all about BOR adjustment).

We agree with the reviewer. 

The phrase "We computed the demand for whole blood or components by extrapolating the sum of whole blood or components demanded by each institution." is not clear. We removed the phrase as the details have been given earlier in the paragraph. 

As demand for blood can occur only in occupied beds of a health care facility, we determined the bed occupancy rate (BOR) adjusted clinical demand per bed. After estimating the total demand in a health care facility, it was adjusted for the actual bed occupancy. We used the total number of beds, bed occupancy rate and the total clinical demand in the health care facilities. 

For example, if the total demand for a 100 bedded hospital is 1000 in a year. The demand per bed per annum is 1000/100 = 10. If the average bed occupancy rate is 80%, the beds used were 80. Therefore, the BOR adjusted clinical demand per bed is 1000/80 = 12.5 units per bed per annum.

3. Please provide in the footnote of all tables the definitions of abbreviations used.

Thanks and we corrected the errors and provided the abbreviations. 

4. In the Table 2, please use the same digit capacity for the "Clinical demand in whole blood units" for the columns related to study data and national estimates.

Thanks for the comment. We have corrected 

5. In the Table 3, please introduce a column with "demand per bed before adjustment" data.

Thanks. The additional columne has been provided.

6. The number of blood transfusions for the Nutritional Anemia is impressive. Could you please comment in the Discussion which haemoglobin level is the indication of blood transfusion for this population?

The details have been incorporated in the discussion section

7. The second most prevalent category requiring blood transfusion is patients with End-stage renal disease. Could you please comment in the Discussion about availability of erythropoetin and parenteral iron medications in India, as well as which haemoglobin level is the indication of blood transfusion in this particular disease? You provided separate estimates for "End-stage renal disease" and "Chronic Kidney Disease" - could you please comment on it?

Your comments have been addressed in the manuscript. 

EPO is available in India but not easily afforded by most of the patients, with a year’s therapy running into several hundreds of thousands of Indian rupees (between 2000 – 5000 USD). The haemoglobin level targeted is above 10g/dL. However, immediate determinants include the availability of blood, and cost. In centres where renal transplant is carried out, constraint is exercised to avoid alloimmunisation for red cells antigens and HLA antigens and where possible leuko-reduced blood is transfused. 

8. In the Table 3 the "Sepsis" and "Disseminated Intravascular Coagulation" are 2 separate categories for adults, but one merged category for paediatric care. Please could you explain why the same questionnaires applied have led to different categories?

On discussion with the experts in the domain (paediatrics and paediatric critical care), it was found that often it was difficult to make a clear-cut difference between the two conditions, especially in critical paediatric patients. Often, the delay in seeking appropriate care would mean the child was septic and progressing to DIC at admission in emergency services. 

For all practice purposes, we clubbed Sepsis in DIC in paediatrics. 

9. In the tables, please use "in thousands" instead of "(in ‘000)".

The tables have been corrected

10. It would be better to use "general medicine" instead of "medical specialty", where appropriate.

We prefer keeping this as “medical speciality” as it is going beyond general medicine (Internal Medicine) as we are considering all sub specialities under this term.

11. While only 30-40% of considered facilities have their own blood bank, could you please describe some details on how the blood or components are distributing from facilities with to those without blood bank? It would be also interesting to know more about blood donation in India - whether donors have some incentives, are there any cultural or religious patterns of blood donation, etc.

Thank you very much for your comment. We highlighted a few aspects in the background section. 

12. Please indicate the volume for the whole blood and components used. Even if their volume per unit is standard, the clear definition is always a good choice.

It is based on 350 ml per single donation which is generally practised in India. We have mentioned it in the method section. 

13. Please avoid abbreviations in the references where possible. Please control for the references completeness, for example "IIPS. National Family Health Survey-2017 (NFHS-4) New Delhi: International Institute for Population Sciences, Ministry of Health and Family Welfare, 2017" would be better to supply with ISBN or web-link.

We have addressed the issues 

14. It is up to your choice, but it would be useful to share the questionnaire in the supplement, to give the opportunity to reproduce the study for other LMICs research groups.

We have included the tools as an attachment. 

5. Reviewers comments to the Author. 

Reviewer #1: 

It’s my opinion that the manuscript is fairly good and well written. The information supplied is also relevant and timely. I have a few questions and some minor concerns. First, the authors should consider the use of a flow chart to describe the sampling strategy. The current approach appears to be long-winded, hence difficult to follow. Further, the authors should have incorporated more references on blood demand from other jurisdictions e.g. regional studies within India or Asia. Although these studies are rare, I’m sure a sufficient number or papers can be found in existing digital repositories. In addition, reference 17 and 22 are similar.

• We have given a flow chart to describe the sampling strategy 

• As you rightly mentioned, there is a dearth of literature related this topic. We tried our best to use the available literature.

• Yes, reference 17 and 22 are similar. We have corrected it. 

Reviewer #2: 

An important topic chosen by a group of authors from different fields, kudos to them. I have few observations which I feel you will be able to answer and further improve the article.

1. Introduction

a. Para 1 last line needs rephrasing

Rephrased

b. para 2 what programmes are being talked about

Voluntary donation and blood safety programmes in countries. Rephrased.

c. Para 3 there is repetition and explanation of the same words, avoid semantics and reduce unnecessary word

Rephrased and added some more details based on editor’s comments 

2. The aim of the study in the abstract and main body is different, please explain.

Rephrased and added some more details based on editor’s comments

3. The study used data from various different institutes, was the ethic clearance from each participating institute taken.

We have not obtained from ethic clearance from each participating institute though we obtained informed consent from all the participating institutes. Moreover, not all participating institution have ethics committee. 

This study is led by the National AIDS Control Organization, Ministry of Health and Family Welfare, Government of India. We obtained clearance from the Ethics committee of the National AIDS Control Organization, Institutional Review Board of Christian Medical College and Hospital, Vellore, India. Besides, this proposal was cleared by the Technical Resource Group (TRG) of Research and Development, National AIDS Control Organization followed by the scientific committee approval from Centres for Disease Control and Prevention, Atlanta.

4. Table one says beds <30, 31-200 etc while text in methods says beds <50 as primary in pvt sector, please justify.

Yes, we agree. Both are different, we provided the bed details to categorise the type of facilities as primary, secondary and tertiary for this study. 

While we provided the type of facilities in Table -1, we also provided a different category based number of beds as well. We have modified the categories now. 

5. Table 4 needs to be made more reader friendly by reducing unnecessary rows.

There are only two empty rows. To have the sub-headings in the same row, we made it like that. 

6. Please explain 'Discussion " para 2 initial 4 lines

Thanks for the comment. We understood the issues and we have modified the paragraph. 

7. Discussion

a) Demand for blood components - has been written and appears out of context. a comment about the clinician's knowledge regarding components was not surveyed and should not be commented upon.

We intend to bring the substantial use of whole blood despite the continuous emphasis on rational use of blood through appropriate component use in India through the National programmes. We discussed the lack of clinician’s knowledge as one of the possible reasons for the extensive use of whole blood in the country. We rephrased the paragraph too. 

b) The discussion needs to be more specific and needs rewriting all over.

We have rewritten/rephrased for better clarity. 

8. Please remove all the unnecessary comments from the conclusion which were not part of the aim & objectives. Please stick to the study per se and not the general comments

We have revised the conclusion section 

Reviewer #3: 

The effort and work are impressive, and it is an interesting descriptive report of the blood supply and usage, but the final estimations are based on many uncertain assumptions. It is not convincingly shown that the selected facilities are representative, the indications for blood transfusions varies and are in many parts not appropriate, according to guidelines in developed countries. The authors make reservations in the discussion and discuss the limitations of the study, but this could be further emphasized.

The study is based on primary data collected from health facilities that transfuse blood and we extrapolated for the country based on the number of beds. Though we do not disagree with the reviewer’s comment about the representation of the study facilities, we would like to emphasise that we followed a sampling strategy that represented all the regions, the different levels of service provision (primary, secondary and tertiary facilities) and type of facilities (public and private) and geographies (rural and urban). 

As suggested we have highlighted the issues in the limitation section. 

Specific comments

Title

I do not fully understand the title: Blood and components? The availability and use of blood components vs whole blood and the demand of different blood components are not analysed.

Something like “The blood supply in India-An estimation of available donors and demand and usage of blood transfusions on a national level”………?

We have titled it as: 

The clinical demand and supply of blood in India: A National level estimation study 

Abstract

line 31 blood and components? Do you mean whole blood and blood components?

Yes, we have corrected

Line 35-36 are the regions, states and facilities all randomly selected? Not chosen as representative for different parts of the country and health care level?

We rephrased it and provided less information due to the word limit. The detailed sampling strategy is in the main text. 

Line 42-44 The study data, from where the national estimations are calculated should be presented.

We have presented the study data in the abstract and the manuscript. 

Line 50-53 The conclusion should be more specific. What is the main message? Some of the findings discussed on page 15, line 282-306 and page 19 line 390-395?

Revised

Introduction

The introduction is informative, describing 2760 blood banks, public or private, often attached to hospitals, component preparation of 53% of the collected blood.

To interpret the data and results a short paragraph is requested of costs in the health care; is the health care free or paid for in public facilities? In private facilities? Are blood transfusions free or paid for? Are blood donors non-remunerated or compensated? Are relatives asked to donate?

We have highlighted a few aspects in the introduction section. 

Methods

Line 92-98. It would be nice with a map of India where you marked the regions and states and the selected facilities.

We have included a map marking the regions and selected states. However, marking the selection of facilities may be difficult with 251 facilities. 

Line 98-100 how was the facilities randomly selected?

The required number of facilities for each strata was randomly selected from the state specific disaggregated list of health facilities. We rephrased the sentence for better clarity. 

Line 98-99 define primary, secondary and tertiary health care facilities, with respect to blood transfusions/blood donations? It is partly explained later in line 108, “based on expert´s consensus”? Do you have a reference, documentation or is it an assumption?

As part of the blood estimation exercise, we conducted Delphi exercise among 50 health care providers from different parts of the country. The expert consensus was based on that. It is included in the manuscript. 

Data collection

Line 133-134 How is the diagnosis and transfusion related information documented? Is it similarly documented in the different facilities?

These are from the hospital records and the diagnosis are similarly documented to a greater extent, since the Blood transfusion services is regulated and licensed. Doubtful information was verified with the clinicians of the respective health care facilities. The transfusion-related information such as the type of components and units were documented similarly. 

Line 136-138 Please describe how you retrieved the data from the systems? Electronically or manually? Do all centers have electronic hospital and blood bank records?

We obtained these data primarily from the manual hospital and blood bank records as all hospitals in our sample maintained manual records 

Line 138 How was the data validated?

The data from the records were checked again for correctness and completeness by another investigator. The doubtful information was verified with the support of the clinicians of the health care facilities. 

Data analysis and computation of clinical demand

Line 140 How was data entered in SPSS?

The data were entered in REDCap application, converted and analysed using SPSS version 24 (IBM, Armonk, NY). We have mentioned it in the manuscript. 

Line 157-158 Are the indications for whole blood and components different or is it a matter of tradition and availability?

It is primary based on availability of components and in some instances, the whole blood was due to non-availability of components and common practice. 

Results

Line 219 Is nutritional anemia an appropriate indication for blood transfusion?

Yes, it is. This is as indicated in the patient records. It is common in India.

Discussion

The discussion includes many interesting aspects. Will the need of blood transfusions increase or decrease in India, referring to international trends? Are blood transfusions used on appropriate indications or is education primary needed?

The demand for blood transfusion indicates a decreasing trend due to increased knowledge and awareness and increasing adherence to principles of patient blood management, though it is difficult to arrive at a conclusion as our study is a cross-sectional one. Certainly, there is a need for education among health care provides to ensure rational and appropriate use of blood in India. This has been emphasised in the discussion and conclusion of the manuscript. 

---

## [Decision Letter · Decision Letter 1]

11 Mar 2022

The clinical demand and supply blood in India: A National level estimation study

PONE-D-21-22354R1

Dear Dr. Mammen,

We’re pleased to inform you that your manuscript has been judged scientifically suitable for publication and will be formally accepted for publication once it meets all outstanding technical requirements.

Kind regards,

Boris Bikbov

Academic Editor

PLOS ONE

Additional Editor Comments (optional):

Reviewers' comments:

Reviewer's Responses to Questions

**Comments to the Author**

1. If the authors have adequately addressed your comments raised in a previous round of review and you feel that this manuscript is now acceptable for publication, you may indicate that here to bypass the “Comments to the Author” section, enter your conflict of interest statement in the “Confidential to Editor” section, and submit your "Accept" recommendation.

Reviewer #1: All comments have been addressed

Reviewer #3: All comments have been addressed

2. Is the manuscript technically sound, and do the data support the conclusions?

Reviewer #1: Yes

Reviewer #3: Yes

3. Has the statistical analysis been performed appropriately and rigorously? 

Reviewer #1: Yes

Reviewer #3: Yes

4. Have the authors made all data underlying the findings in their manuscript fully available?

Reviewer #1: Yes

Reviewer #3: Yes

5. Is the manuscript presented in an intelligible fashion and written in standard English?

Reviewer #1: Yes

Reviewer #3: Yes

6. Review Comments to the Author

Reviewer #1: (No Response)

Reviewer #3: I find that that the authors have addressed the comments and objections appropriately. They have clarified the issues with inclusion of centers, indication for blood transfusions, calculations, etc. I think the manuscript gives interesting information of blood supply and demand in India, and the information may also be representative for other countries. I appreciate the added limitations of the study, and requirement of education and implementation of other treatment strategies in nutritional anemia.

7. PLOS authors have the option to publish the peer review history of their article (what does this mean?). If published, this will include your full peer review and any attached files.

Reviewer #1: No

Reviewer #3: **Yes: **Agneta Wikman, Professor, Senior Consultant, Transfusion Medicin, Karolinska Institutet

---

## [Editor Report · Acceptance letter]

28 Mar 2022

PONE-D-21-22354R1 

The clinical demand and supply of blood in India: A National level estimation study 

Dear Dr. Mammen:

I'm pleased to inform you that your manuscript has been deemed suitable for publication in PLOS ONE. Congratulations! Your manuscript is now with our production department. 

Kind regards, 

on behalf of

Dr. Boris Bikbov 

Academic Editor

PLOS ONE